# A Critical Look at Classic Test-Time Adaptation Methods in Semantic Segmentation

## Abstract

Test-time adaptation (TTA) aims to adapt a model, initially trained on training data, to potential distribution shifts in the test data. Most existing TTA studies, however, focus on classification tasks, leaving a notable gap in the exploration of TTA for semantic segmentation. This pronounced emphasis on classification might lead numerous newcomers and engineers to mistakenly assume that classic TTA methods designed for classification can be directly applied to segmentation. Nonetheless, this assumption remains unverified, posing an open question. To address this, we conduct a systematic, empirical study to disclose the unique challenges of segmentation TTA, and to determine whether classic TTA strategies can effectively address this task. Our comprehensive results have led to three key observations. First, the classic batch norm updating strategy, commonly used in classification TTA, only brings slight performance improvement, and in some cases it might even adversely affect the results. Even with the application of advanced distribution estimation techniques like batch renormalization, the problem remains unresolved. Second, the teacher-student scheme does enhance training stability for segmentation TTA in the presence of noisy pseudo-labels. However, it cannot directly result in performance improvement compared to the original model without TTA. Third, segmentation TTA suffers a severe long-tailed imbalance problem, which is substantially more complex than that in TTA for classification. This long-tailed challenge significantly affects segmentation TTA performance, even when the accuracy of pseudo-labels is high. In light of these observations, we conclude that TTA for segmentation presents significant challenges, and simply using classic TTA methods cannot address this problem well. Therefore, we hope the community can give more attention to this challenging, yet important, segmentation TTA task in the future. The source code will be publicly available.

## 1 Introduction

Test-time adaptation (TTA) focuses on tailoring a pre-trained model to better align with unlabeled test data at test time (Sun et al., 2020). This paradigm is popular since the test data may unavoidably encounter corruptions or variations, such as Gaussian noise, weather changes, and many other reasons (Hendrycks & Dietterich, 2019; Koh et al., 2021). Furthermore, the training and test data can not co-exist due to privacy concerns. These challenges have propelled TTA to the forefront as an emergent and swiftly evolving paradigm (Sun et al., 2020; Wang et al., 2021; Niu et al., 2022b; 2023; Liang et al., 2023).

Recently, an array of test-time adaptation (TTA) techniques (Sun et al., 2020; Wang et al., 2021) have emerged for classification problems. Broadly, these can be classified into two main categories: Test-Time Training (TTT) (Sun et al., 2020; Liu et al., 2021) and Fully TTA (Wang et al., 2021; Niu et al., 2022a). Compared to test-time training (TTT), fully TTA (TTA for short) is more practical, since TTT needs to change the original model training which may be infeasible due to privacy concerns. The key idea of TTA methods is to define a proxy objective at test time to adapt the pre-trained model in an unsupervised manner. Typical proxy objectives include entropy minimization (Wang et al., 2021), pseudo labeling (Liang et al., 2020) and class prototypes (Su et al., 2022).

While the majority of TTA studies have centered on classification problems, real-world scenarios frequently highlight the ubiquity and critical nature of semantic segmentation tasks. A prime in-

stance is autonomous driving, where systems must accurately and instantaneously segment an array of dynamic and unpredictable perceptions (Li et al., 2023). Thus, even though classic TTA techniques offer marked advantages for classification, their suitability and performance for semantic segmentation remain unknown and largely under-explored.

To fill this gap, we seek to attain a comprehensive understanding of the current classic TTA for semantic segmentation and to identify critical, unresolved issues warranting further exploration. Our analyses provide three key observations in TTA for semantic segmentation. First, the classic batch norm updating strategy, prevalent in classification TTA, offers marginal performance gains and can sometimes even deteriorate the outcomes. Advanced techniques like batch renormalization and large batch sizes fail to address this limitation effectively. Secondly, while the teacher-student approach bolsters training stability in segmentation TTA amidst noisy pseudo-labels, it does not necessarily elevate the performance beyond models not employing TTA. Lastly, segmentation TTA grapples with an acute long-tailed imbalance issue, which is more intricate than its counterpart in classification TTA. This long-tailed dilemma profoundly impedes segmentation TTA efficacy, even with high-accuracy pseudo-labels. These insights underline the critical nuances and challenges in employing classic TTA strategies in semantic segmentation scenarios.

**Main contributions**. To the best of our knowledge, this paper is among the first to comprehensively investigate test-time adaptation for semantic segmentation, which is important yet under-explored. Our main observations are summarized as follows: 1) Batch norm update techniques are unable to handle TTA for semantic segmentation. 2) Teacher-student scheme helps stabilize segmentation TTA, particularly when faced with substantial pseudo-label noise. 3) Long-tailed class imbalance presents a profound challenge in segmentation TTA, and test-time augmentation only partially relieves long-tailed biases in segmentation TTA. We hope that our detailed analyses and comprehensive experimental results can provide insights and a broader understanding to the community.

## 2 PROBLEM STATEMENT AND EXPERIMENTAL SETUPS

**Problem statement**. This paper focuses on test-time adaptation (TTA) for semantic segmentation. Let $\mathcal{D}^{train} = \{(\mathbf{x}_i, \mathbf{y}_i)\}_{i=1}^{N} \in \mathcal{P}^{train}$ be the training data, where $\mathbf{x}$, $\mathbf{y}$ and $N$ represent the data, labels and data amounts, respectively. Let $f_\Theta(\mathbf{x})$ denote a pre-trained segmentation model with parameters $\Theta$. The goal of segmentation TTA is to adapt $f_\Theta(\mathbf{x})$ to the unlabeled test data $\mathcal{D}^{test} = \{\mathbf{x}_i\}_{i=1}^{M} \in \mathcal{P}^{test}$ with different data distributions, i.e., $\mathcal{P}^{train}(\mathbf{x}) \neq \mathcal{P}^{test}(\mathbf{x})$. Under the TTA scheme (Wang et al., 2021), the model $f_\Theta(\mathbf{x})$ receives a batch of test data at each time step and will be updated in an unsupervised manner at test time.

**Classic TTA strategies**. In this paper, our primary objective is to uncover the unique challenges posed by segmentation TTA and investigate whether classic TTA strategies can effectively address them. To achieve this, we delve into several well-established strategies, including batch norm updating (Zhao et al., 2023a), teacher-student scheme (Wang et al., 2022), and pseudo labeling (Zhang et al., 2023b), all of which have demonstrated effectiveness in classification TTA.

**Experimental setups**. To ensure consistent evaluations of various TTA approaches, we conduct empirical studies based on several widely used semantic segmentation datasets, including ACDC (Sakaridis et al., 2021), Cityscapes-foggy (CS-fog) (Sakaridis et al., 2018) and Cityscapes-rainy (CS-rain) (Hu et al., 2019). In addition, we strictly follow the implementation details outlined in previous studies (Wang et al., 2022; Botet Colomer et al., 2023), and use Segformer-B5 (Xie et al., 2021) as the pre-trained model. Unless otherwise specified, all experiments are conducted with a batch size (BS) of 1, mirroring real-world scenarios where the test data often arrives one by one in a sequential manner.

## 3 DOES BATCH NORM UPDATING WORK FOR SEGMENTATION TTA?

### 3.1 BATCH NORM UPDATING FAILS IN SEGMENTATION TTA

We start with batch normalization (BN) updating strategies (Nado et al., 2020; Schneider et al., 2020). Most existing BN-based TTA methods (Wang et al., 2021; Niu et al., 2022b), contrary to typical deep learning pipelines, compute distribution statistics directly from test data, rather than

Table 1: Accuracy of batch norm updating strategies (i.e., TENT (Wang et al., 2021) and its variants) on ACDC, Cityscapes-fog and Cityscapes-rain. SO indicates the *source only* model without test-time adaptation, while BS indicates the batch size of test data at each iteration. Except that the TENT (larger BS) variant uses a batch size of 4, all TENT methods are based on BS = 1 as mentioned in Section 2.

| Method | A-fog | A-night | A-rain | A-snow | CS-fog | CS-rain | Avg |
|---|---|---|---|---|---|---|---|
| SO | 69.1 | 40.3 | 59.7 | 57.8 | 74.2 | 66.6 | 61.3 |
| TENT | 64.2 (-4.9) | 40.0 (-0.3) | 57.6 (-2.1) | 55.1 (-2.7) | 73.9 (-0.3) | 66.8 (+0.2) | 59.1 (-2.2) |
| TENT (larger BS) | 65.3 (-3.8) | 40.6 (+0.3) | 57.3 (-2.4) | 54.2 (-3.6) | 71.6 (-2.6) | 66.7 (+0.1) | 59.3 (-2.0) |
| TENT (BN-fixed) | 69.0 (-0.1) | 40.2 (-0.1) | 60.1 (+0.4) | 57.3 (-0.5) | 74.1 (-0.1) | 66.5 (-0.1) | 61.2 (-0.1) |

starting with or inheriting those from the training data. These methods only update the batch normalization layers during TTA, restricting changes exclusively to the model parameters. This ensures that the core learned features remain intact, while only the normalization gets adjusted based on the test data. While these approaches have demonstrated their effectiveness in bridging domain gaps for image classification at test time, their efficacy in semantic segmentation is yet to be thoroughly explored and validated.

To delve deeper into this, we conduct a thorough evaluation of BN-based TTA methods in semantic segmentation based on a classic method (TENT (Wang et al., 2021)). Specifically, TENT adapts models by using the BN statistics from mini-batch test data (with a BS = 1 in TTA segmentation) instead of those inherited from the training data, and updating the affine parameters of BS through entropy minimization. Moreover, we explore two variants of TENT: (1) TENT (larger BS) seeks to enhance TENT's performance by utilizing a larger batch size of 4, aiming for a more precise estimation of distribution statistics. (2) TENT (BN-fixed) retains the BN statistics from the training data without adaptation and solely updates the affine parameters of BS through entropy minimization.

As shown in Table 1, we have three main observations. First, all TENT variants perform worse than *Source Only* (SO) baseline, highlighting the difficulties that classic batch norm updating methods encounter in TTA segmentation. Second, even though utilizing a larger batch size marginally elevates TENT's performance, it remains overshadowed by SO. Last, the TENT (source) variant, although the affine parameters of BN are updated, achieves performance only similar to SO. This shows that retaining the BN statistics from the training data performs a key factor, while updating the affine parameters of BN does not bring the expected improvement. In summary, batch norm updating strategies, despite performing well in classification TTA, do not meet anticipated outcomes in semantic segmentation TTA. Please refer to Section 3.3 for more discussions on distribution estimation tricks like larger batch size and batch renormalization.

## 3.2 ALIGNING BATCH NORM STATISTICS LOSES ITS MAGIC IN SEGMENTATION

We next aim to probe the underlying reasons for the poor performance of BN-based TTA methods in semantic segmentation. Before diving into this detailed analysis, we first provide a foundational overview of BN updating to ensure clarity and comprehension. Let $f \in \mathbb{R}^{B \times C \times H' \times W'}$ represent a mini-batch of features, where $C$ indicates channel numbers, $H'$ is the height of features, and $W'$ is the width. BN normalizes $f$ using the the distribution statistics of mean $\mu$ and variance $\sigma$ (both $\mu$ and $\sigma$ belong to $\mathbb{R}^C$). The normalization is mathematically expressed as:

$$f^* = \gamma \cdot f' + \beta, \quad where \quad f' = \frac{f - \mu}{\sigma}, \tag{1}$$

where $\gamma, \beta \in \mathbb{R}^C$ are learnable affine parameters of BN that represent scale and shift, respectively. During inference, $\mu, \sigma$ are set to $\mu^{ema}, \sigma^{ema}$, which are the exponential-moving-average (EMA) estimation of distribution statistics. Previous BN-based TTA methods for classification have shown that in situations where there is a distribution shift between training and test data, i.e., $\mathcal{P}^{train}(\mathbf{x}) \neq \mathcal{P}^{test}(\mathbf{x})$, replacing the EMA estimation $\mu^{ema}, \sigma^{ema}$ with the test mini-batch statistics can boost the model performance (Wang et al., 2021) when the test mini-batch statistics are accurate.

However, Table 1 has demonstrated that such a strategy does not make sense in semantic segmentation. The challenges arise from the model's difficulty in accurately assessing the test data statistics during adaptation for segmentation. To shed light on this, we visualize the estimated distribution

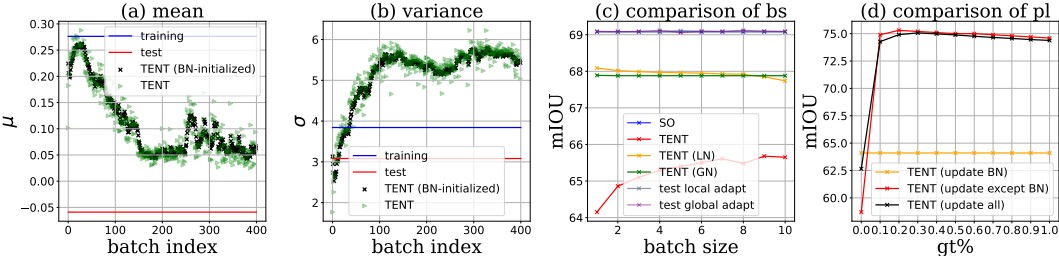

Figure 1: Quantitative metrics analysis. (a) and (b) capture the BN distribution statistics throughout online adaptation. (c) shows the differential impacts of batch norm updating across different batch sizes. (d) delves into the effects of varying updating strategies based on TENT, contrasting different proportions of pseudo-labels with the rest being ground-truth labels.

statistics of BN in Figure 1 (a)-(b). To be specific, we train the model from scratch on both the training Cityscapes data and the test ACDC-fog data, followed by recording the BN distribution statistics, represented by "training" (the blue line) and "test" (the red line) in Figure 1 (a)-(b). Subsequently, we employ the aforementioned TENT to adapt the trained model to the test data, and record the change of BN distribution statistics. Specifically, TENT adjusts the BS statistics based solely on the mini-batch test data independently at each iteration. In contrast, TENT (BN-initialized) starts with the BN distribution statistics from the training data model and progressively adapts the BN statistics using EMA, instead of computing statistics independently for each mini-batch.

Figure 1 (a)-(b) leads to four main findings. First, the distributional discrepancy between the "training" and "test" data is pronounced. Second, while the TENT (BN-initialized) — represented by the black dots in Figure 1 (a)-(b) — does endeavor to adjust to the test data, it fails to estimate test data very well, still remaining misalignment relative to the true test data distribution. Thirdly, the BN statistics' evolution in TENT (depicted by the green points) mirrors that of TENT (BN-initialized) quite closely. This resemblance arises because, even though TENT's BN statistics are not inherited and are recalibrated based on individual mini-batches of test data at every iteration, the rest of the model parameters are indeed derived from the training data model. Consequently, the initial feature distribution still aligns more closely with the training data's distributional characteristics, preventing direct approximation of the test data distribution. As adaptation progresses, while there is a trend towards aligning with the test distribution, it, much like TENT (BN-initialized), ultimately fails to capture it accurately. Lastly, we notice a pronounced increase in variance, indicating a widening divergence in the distribution estimation. In summary, the imprecise estimation of the test data distribution renders BN updating strategies ineffective for TTA segmentation, with the escalating variance even potentially imparting detrimental effects on the model performance.

### 3.3 DISTRIBUTION ESTIMATION TRICKS CANNOT RESOLVE THE PROBLEM

In light of the above discussions, we next ask whether further using distribution estimation tricks can rectify the issues associated with the distribution estimation of BN updating in segmentation TTA. In response, we investigate three approaches: harnessing a larger batch size, adopting batch renormalization, and leveraging ground-truth labels (mainly for empirical analysis).

**Larger batch size**. Previous studies (Niu et al., 2023) have shown that using a larger batch size can enhance the BN updating for classification TTA. Driven by this rationale, we investigate the impact of different batch sizes (ranging from 1 to 10) on segmentation TTA, where we also provide the results based on layer normalization (LN) (Ba et al., 2016) and group normalization (GN) (Wu & He, 2018). As shown in Figure 1 (c), an increase in batch size does indeed enhance BN updating. However, this enhancement does not translate to an improvement over SO (i.e., the training data model without adaptation). This indicates that merely increasing the batch size cannot adequately solve the issue of normalization-based TTA methods.

**Batch renormalization**. Utilizing local test mini-batch statistics for model adaptation proves unreliable, especially when confronting persistent distribution shifts (Yuan et al., 2023). Such unreliability originates from error gradients and imprecise estimations of test data statistics. In response, we delve into two test-time batch renormalization techniques (Zhao et al., 2023a; Yuan et al., 2023), namely *Test Local Adapt* and *Test Global Adapt*, aiming to refine distribution estimation. The for-

Table 2: Results of the teacher-student scheme on ACDC (%). "SO"/"Single"/"TS" are short for source only/the single model/the teacher-student scheme, and "PL"/"Aug" are short for pseudo-labeling/test-time augmentation, respectively.

| Method | PL | Aug | A-fog | A-night | A-rain | A-snow | Aug |
|--------|----|----|-------|---------|--------|--------|-----|
| SO | | | 69.1 | 40.3 | 59.7 | 57.8 | 56.7 |
| Single | ✓ | | 55.5 (-13.6) | 29.8 (-10.5) | 45.5 (-14.2) | 41.4 (-16.4) | 43.1 (-13.7) |
| TS | ✓ | | 68.3 (-0.8) | 39.5 (-0.8) | 59.8 (+0.1) | 57.4 (-0.4) | 56.3 (-0.4) |
| Single | ✓ | ✓ | 56.1 (-13.0) | 30.0 (-10.3) | 45.8 (-13.9) | 41.2 (-16.6) | 43.3 (-13.4) |
| TS | ✓ | ✓ | 69.9 (+0.8) | 40.4 (+0.1) | 61.8 (+2.1) | 59.0 (+1.2) | 57.8 (+1.1) |

mer, *Test Local Adapt*, leverages source statistics to recalibrate the mini-batch test data distribution estimation, whereas *Test Global Adapt* uses test-time moving averages to recalibrate the overall test distribution estimation. As shown in Figure 1 (c), while batch renormalization strategies do enhance the performance of TENT, their performance is just comparable to that of SO and cannot lead to performance improvement.

**Ground-truth labels**. To analyze the impact of pseudo-label noise on distribution estimation, we leverage true labels for empirical studies. Moreover, to analyze the effects of updating different network components, we further explore three distinct updating strategies. (1) TNET (update BN): the affine parameters in BN are updated; (2) TNET (update except BN) involves updating parameters except for BN; (3) TNET (update all): all model parameters are updated. As shown in Figure 1(d), when solely relying on pseudo-labels, TENT (update BN) outperforms its counterparts due to its minimal parameter updating, making it less susceptible to the noise of pseudo-labels. In contrast, the other baselines exhibit markedly inferior performance under these conditions. However, as the quality of pseudo-labels improves—with the incorporation of more ground truth labels—there's a significant performance boost in TENT (update expect BN) and TENT (update all). Yet, TENT (update BN) remains stagnant, not showing the same enhancement. This further demonstrates the limitations of existing BN updating TTA strategies in semantic segmentation.

## 4 DOES THE TEACHER-STUDENT SCHEME WORK FOR SEGMENTATION TTA?

### 4.1 TEACHER-STUDENT SCHEME HELPS STABILIZE SEGMENTATION TTA

The teacher-student exponential moving average (TS-EMA) scheme (Hinton et al., 2015) has been shown to enhance model training and accuracy (Tarvainen & Valpola, 2017). Many recent methods (Wang et al., 2022; Yuan et al., 2023; Tomar et al., 2023) have introduced it into TTA by using a weighted-average teacher model to improve predictions. The underlying belief is that the mean teacher's predictions are better than those from standard, single models. However, the precise influence of TS-EMA on segmentation TTA has not been thoroughly investigated. In this section, we seek to delve into its empirical impact. For the implementation of the TS-EMA scheme, we follow CoTTA (Wang et al., 2022) to update the student model by $\mathcal{L}_{PL}(\mathbf{x}_{\mathcal{T}}) = -\frac{1}{C}\sum_c^C \tilde{y}_c \log \hat{y}_c$, where $\tilde{y}_c$ is the probability of class $c$ in the teacher model's soft pseudo-labels prediction, and $\hat{y}_c$ is the output of the student model.

To figure out whether the TS-EMA scheme indeed stabilizes TTA for semantic segmentation, we compare the TS-EMA scheme and the single-model scheme (Single) with pseudo-labeling (PL) and test-time augmentation (Aug) (Lyzhov et al., 2020), to comprehensively contrast their performance. As shown in Table 2, the single-model scheme consistently underperforms compared to the SO baseline, a trend that persists even with the integration of PL and Aug. In stark contrast, the TS-EMA scheme maintains relatively stable performance. Using PL, it experiences only minor drops in categories like "A-fog" and "A-night", and even shows an improvement in "A-rain". Moreover, when employing both PL and Aug, TS outperforms the SO baseline. In light of these observations, we conclude that TS-EMA stands out as a robust method to improve the training stability of TTA.

Table 3: Comparison between the teacher-student scheme and the source-only model on ACDC (%). "SO"/"TS" are short for source only/the teacher-student scheme, and "PL"/"Aug" are short for pseudo-labeling/test-time augmentation, respectively.

| Method | PL | Aug | A-fog | A-night | A-rain | A-snow | Aug |
|--------|-----|------|--------|----------|---------|---------|-----|
| SO | | | 69.1 | 40.3 | 59.7 | 57.8 | 56.7 |
| SO | | ✓ | 70.9 (+1.8) | 41.2 (+0.9) | 62.3 (+2.6) | 59.6 (+1.8) | 58.5 (+1.8) |
| TS | ✓ | ✓ | 69.9 (+0.8) | 40.4 (+0.1) | 61.8 (+2.1) | 59.0 (+1.2) | 57.8 (+1.1) |

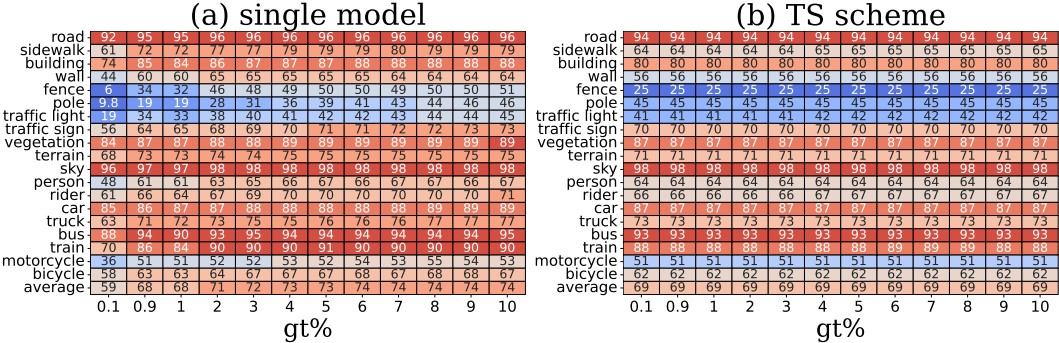

Figure 2: Comparison with the single-model scheme and the teacher-student scheme under different degrees of pseudo-labels. As the accuracy of pseudo-labels increases, the performance of the single model experiences continual enhancement to 74%. However, the teacher-student model's performance remains stagnant at 69%.

## 4.2 TEACHER-STUDENT SCHEME IS NOT DEFINITELY USEFUL FOR SEGMENTATION TTA

While previous sections attest to the efficacy of the TS-EMA scheme, a closer examination of Table 3 underscores a notable observation: when the SO baseline is fortified with test-time augmentation, its performance surpasses that of the TS combined with both PL and Aug. This suggests that the primary advantage of TS-EMA may lie in mitigating the noise introduced by PL, thereby allowing Aug to function more effectively.

This finding provokes a subsequent question: if the accuracy of the pseudo-labels is enhanced, would the TS model also exhibit improved performance as shown in previous studies (Tarvainen & Valpola, 2017)? To answer this question, we adjust the experimental setting, concentrating on situations where pseudo-labels become increasingly accurate, marked by a growing proportion of ground-truth labels. In this context, we assume access to these ground-truth labels so that we can empirically assess model performance across varying pseudo-label accuracies.

We continue our comparison between the single-model scheme and the teacher-student scheme. As shown in Figure 2, we have plotted the IoU (Intersection over Union) metrics for each class against varying levels of ground-truth (GT). This visualization helps us critically assess how the performance trajectory of these two schemes adjusts as the accuracy of the pseudo-labels becomes more accurate

Upon a detailed observation, it becomes evident that both the single-model and teacher-student schemes exhibit similar performance trends. When the precision of the pseudo-labels hits an approximate threshold of 1%[1], the single-model scheme achieves a performance that is almost neck-and-neck with that of the teacher-student scheme.

However, as we progress beyond this pseudo-label precision threshold, an interesting divergence arises: while the single model continues to better its performance, the teacher-student model appears to stagnate. Its mIoU metric remains static at 0.69. In stark contrast, the single model exhibits a commendable improvement, witnessing its mIoU metric jump from an initial 0.59 to a robust 0.74.

Given this observation, one could infer a potential limitation intrinsic to the teacher-student scheme. Despite having increasingly accurate pseudo-labels at its disposal, it does not exhibit the expected adaptability and responsiveness, unlike its single-model counterpart.

---

[1]To put this into perspective, for an ACDC image, 0.01% GT translates to a total of $0.01 * 1080 * 1920 = 22572$ pixels.

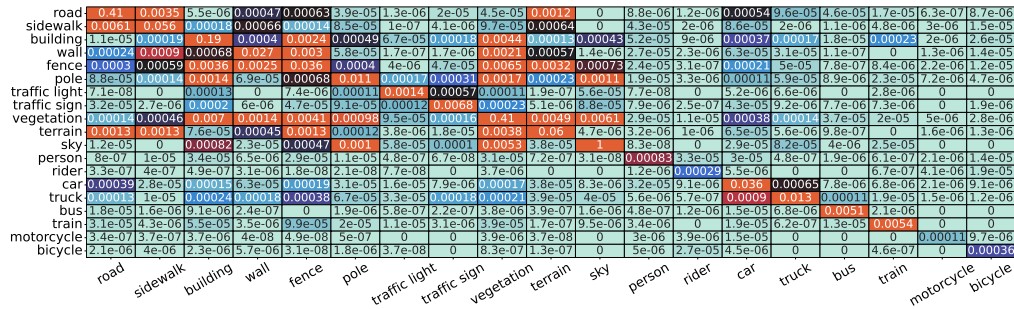

Figure 3: Confusion matrix of ACDC-fog. Here, x-axis indicates the predicted labels, while y-axis represents the ground-truth labels, and the data has been normalized to Min-Max Normalization. We observe a substantial disparity in performance between the majority and minority classes, underscoring the challenges inherent in semantic segmentation.

## 5 DOES CLASS IMBALANCE INFLUENCE SEGMENTATION TTA?

### 5.1 SEGMENTATION TTA SUFFERS SEVERE LONG-TAILED PROBLEM

Semantic segmentation inherently grapples with the challenge posed by data imbalance (Hoyer et al., 2022; Zhang et al., 2023a). Certain semantic classes, such as sky and buildings, are predisposed to occupy vast areas populated with significantly more pixels, often leading them to dominate the visual space, prevalent in numerous realistic pixel-level classification endeavors.

When placed in the context of test-time adaptation, the long-tailed (LT) problem becomes more pronounced, manifesting as an obvious bias in test-time optimization towards dominant classes (Zhao et al., 2023a; Zhang et al., 2022). As shown in Figure 4, the numerical disparity between the majority and minority classes surpasses a staggering 1000-fold difference. This stark contrast is evident when compared to common datasets used in classification tasks, such as CIFAR10-LT, where the most majority class is only in the thousand-level range and has $100\times$ more samples than the most minority class (Wei et al., 2021). Adding to the challenge is the nature of semantic segmentation itself, which involves copious pixel-level labels, further complicating the LT complexity. In this section, we aim to shed light on the challenges of the LT problem as it manifests in segmentation TTA.

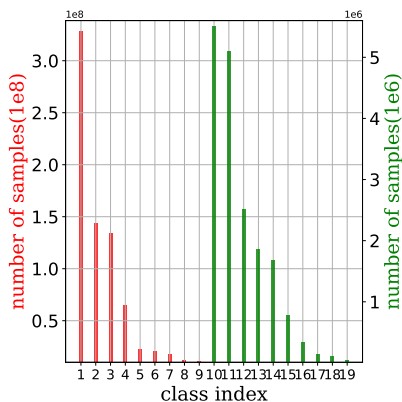

Figure 4: The class distribution in ACDC-fog is highly imbalanced, where the order of magnitude for classes 1 to 9 exceeds 1e8, where that for classes 10 to 19 exceeds 1e6.

### 5.2 LONG-TAILED PROBLEM IN SEGMENTATION IS MORE COMPLEX THAN CLASSIFICATION

We then show the intricate complexity and challenge inherent in semantic segmentation, making it markedly more difficult than classification tasks. To delve deeper into this issue, we assume that the model can generate high-confidence pseudo-labels for test data during adaptation and subsequently analyze the resultant state of the model.

The confusion matrix of ACDC-fog is displayed in Figure 3, unveiling extreme variations in the outcomes for each class, reflecting the substantial discrepancy in the metric across different classes. For example, when a pixel is predicted to be *fence*, its true labels—rider, motorcycle, and bicycle—are 6, 16, and 10, respectively, contrasting sharply with other classes that are in the tens of thousands. We suggest this stark difference elucidates the extreme variation and irregularity in the model's predictive accuracy for different classes.

To further detailed analysis of LT, we also show the quantitative metrics of each class on ACDC-fog[2], as shown in Figure 5. We conduct a comparison of the results between two experiments:

---

[2]The results on the other domains of ACDC are reported in Figure 6-Figure 8 (cf. Appendix A).

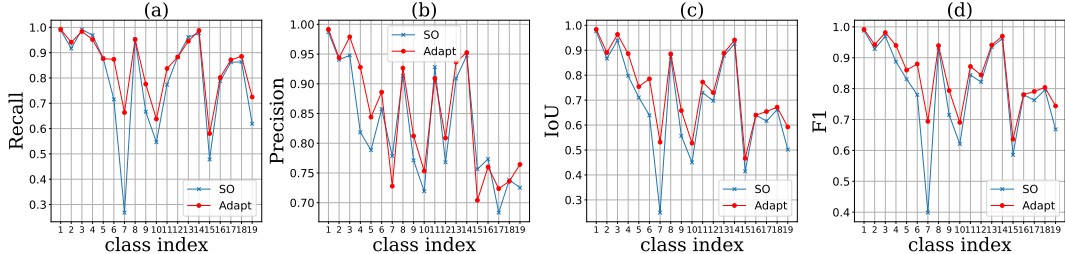

Figure 5: Quantitative metrics analysis (ACDC-fog). After adaptation, the IoU and F1 score for the majority of classes experience improvement. Specifically, there is an increase in the Recall for numerous classes, while the Precision for a limited number of classes actually witnesses a decline.

Table 4: Ablation studies on ACDC-fog of data augmentation (Aug) on in terms of F1 score and mIoU.

| Method | Aug | F1 | | | | mIoU | | | |
|---|---|---|---|---|---|---|---|---|---|
| | | head | mid | tail | Avg | head | mid | tail | Avg |
| Pseudo-labeling | | 89.8 | 82.4 | 82.7 | 85.6 | 82.8 | 71.1 | 69.9 | 74.5 |
| | ✓ | 89.7 (-0.1) | 82.7 (+0.3) | 81.6 (-1.1) | 84.7 (-0.9) | 82.9 (+0.1) | 73.5 (+2.4) | 74.3 (+4.4) | 76.9 (+2.4) |

*Source Only* (SO) and *Adapt* (where we fine-tuned the source model using 100% ground-truth labels). Firstly, as evident in all the plots of Figure 5, the majority classes consistently achieve exceptionally high scores across all metrics, whereas the minority classes do not consistently perform the worst. Secondly, following the adaptation process (involving the addition of supervised information to model training), the recall of most classes shows improvement, while the precision of certain minority classes experiences a decrease. For instance, after adaptation, the Recall of class 7 increases from 0.27 to 0.68, while the precision decreases from 0.78 to 0.73. An increase in Recall alongside a decrease in precision implies a reduction in False Negative and an increase in False Positive. This indicates that the model is less likely to miss pixels of this class (predicting it as other classes) while becoming more prone to predicting instances of other classes as this class. This phenomenon diverges from the patterns observed in classification tasks (Wei et al., 2021) and does not align with conventional wisdom, adding complexity to the uncovering of underlying patterns.

Although conventional wisdom may suggest that the performance of majority classes surpasses that of minority classes, we observe that this rule does not hold true in segmentation tasks. For example, in the third plot of Figure 5, class 19 attains an IoU of 0.59, whereas class 7 achieves an IoU of 0.52. However, it is worth noting that the count of class 7 is $10^6$ while the count of class 14 is $10^5$, as illustrated in Figure 4. In summary, a segmentation task in TTA proves to be significantly more intricate than a classification task, and the reason might be that the long-tailed phenomenon may cause error accumulation at the pixel level and negatively affect the training process.

## 5.3 AUGMENTATION PARTIALLY RELIEVES LONG-TAILED BIASES IN SEGMENTATION TTA

Having already identified the long-tailed problem as a key challenge in segmentation TTA, and considering the effectiveness of test-time augmentation (cf. Table 3), we ponder the potential of test-time augmentation to alleviate the issue of tail-class information scarcity. Following this, we conduct an ablation study for test-time augmentation in terms of the F1 Score and mIoU.

As shown in Table 4, we find that employing data augmentation results in a 2.4 increase in mIou; however, it simultaneously leads to a 0.9 decrease in the F1 Score. This suggests that the model, post-augmentation, intensifies its prediction of minority classes, leading to a simultaneous rise in both True Positive and False Positive, thereby boosting mIoU. Nonetheless, the nuanced balance of Recall and Precision in the F1 Score leads to a less pronounced change. Regarding the tail classes, we observe a notable 4.4% increase in mIoU, contrasted by a 1.1% decline in F1. This showcases that while augmentation enhances the model's detection of tail classes, it does not uniformly improve its precision for these classes. In light of the above observations, we conclude that test-time augmentation partially relieves long-tailed biases in segmentation TTA.

## 6 RELATED STUDIES

**Test-time adaptation**. Generally, the purpose of a typical TTA task is classification, regression or segmentation. Existing works mainly focus on classification. TENT (Wang et al., 2021) adapts batch normalization layers based on entropy minimization, i.e., the confidence of the target model is measured by the entropy of its predictions. Since performing adaptation for all test samples is computationally expensive, EATA (Niu et al., 2022b) actively selects reliable samples to minimize the entropy loss during inference. Furthermore, it also introduces a Fisher regularizer to filter out redundant samples to reduce computational time. SAR (Niu et al., 2023) is a reliable and sharpness-aware entropy minimization approach that can suppress the effect of noisy test samples with large gradients. Thus, it can stabilize online TTA under wild settings such as small batch, mixed shifts and imbalanced label shifts. This model-free paradigm can avoid error accumulation and catastrophic forgetting problems in image classification.

Besides entropy-based approaches, many other strategies are also introduced to address TTA. Ada-Contrast (Chen et al., 2022) combines contrastive learning and pseudo labeling to handle TTA, where better target representations can be learned in a contrastive manner and the pseudo-labels can be refined. In real-world applications, the perception system of a machine is running in continuously changing environments, where the data distribution varies from time to time. AdaNPC (Zhang et al., 2023b) is a parameter-free TTA approach based on a K-Nearest Neighbor (KNN) classifier, where the voting mechanism is used to attach labels based on $k$ nearest samples from the memory. Different from traditional continual TTA approaches, CTTA-VDP (Gan et al., 2023) introduces a homeostasis-based prompt adaptation strategy that freezes the source model parameters during the continual TTA process. TTAB (Zhao et al., 2023b) unveils three common pitfalls in prior TTA approaches under classification tasks, based on a large-scale open-sourced benchmark and thorough quantitative analysis. Similar to classification problems, only one label is attached to a sample in a regression task. However, the cross-entropy loss, which is effectively used in classification, is inherently inapplicable to a regression problem such as pose estimation (Li et al., 2021).

**Semantic segmentation**. Currently, only a few works attempt to address segmentation TTA. HAMLET (Botet Colomer et al., 2023) can handle unforeseen continuous domain changes, since it combines a specialized domain-shift detector and a hardware-aware backpropagation orchestrator to actively control the model's real-time adaptation for semantic segmentation. CoTTA (Wang et al., 2022) can tackle these issues in terms of two aspects. Firstly, it reduces error accumulation based on weight-averaged and augmentation-averaged predictions. Secondly, it avoids catastrophic forgetting by stochastically restoring a small part of the source model's weights. Segmentation tasks are also pervasive in medical images, since the scanner model and the protocol differ across different hospitals. This issue can be handled by introducing an adaptable per-image normalization module and denoising autoencoders to incentivize plausible segmentation predictions (Karani et al., 2021). Similar to (Zhao et al., 2023b) which systematically evaluates the strengths and limitations of existing TTA approaches under classification, the segmentation community of TTA also needs insightful guidelines to assist both the academic community and industry practitioners.

## 7 CONCLUSIONS

Test-time adaptation (TTA) has emerged as a promising and practical research field to attack the robustness challenge under distribution shifts. Effective segmentation approaches would bring great convenience in tasks such as autonomous driving, while most existing TTA approaches are tailored for classification problems. Our purpose is to assist both experienced researchers and newcomers in better understanding TTA for semantic segmentation. In this paper, we present systematic studies and comprehensive analyses to investigate the applicability of classic TTA strategies on segmentation tasks. Extensive experimental results indicate that the effectively used TTA strategies, such as batch norm updating and teacher-student EMA scheme, do not work well in segmentation tasks. Moreover, we also analyze the pitfalls and possible solutions to improve the performance. We hope that more researchers can join the TTA community and build a common practice for segmentation.

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

# A  MORE RESULTS REGARDING LONG-TAILED PROBLEMS

We provide more results on the night, rain and snow domains within ACDC, which further shows the the complexity of long-tailed problems in semantic segmentation.

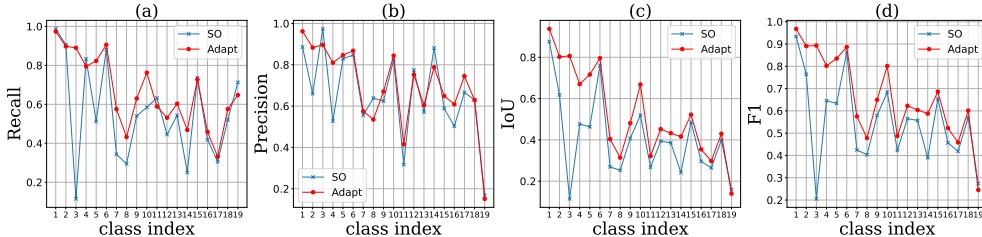

Figure 6: Quantitative metrics analysis on ACDC-night.

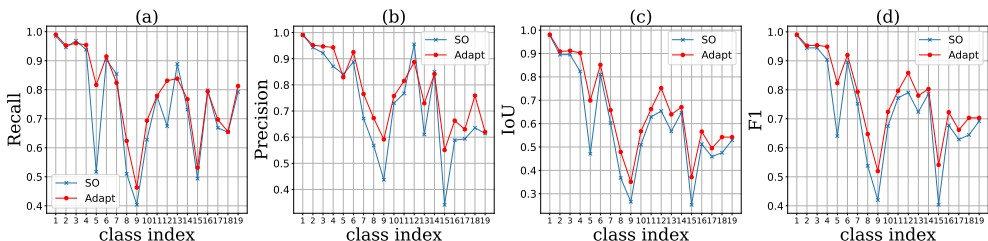

Figure 7: Quantitative metrics analysis on ACDC-rain.

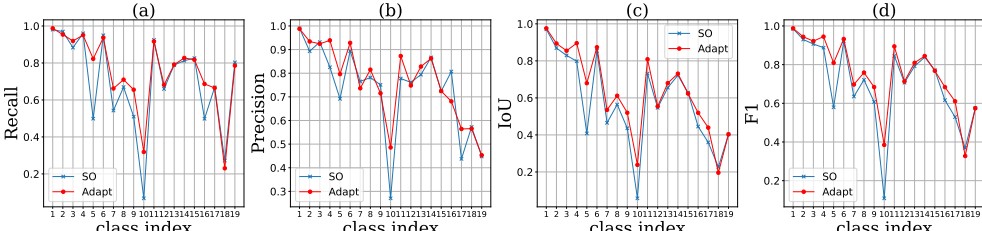

Figure 8: Quantitative metrics analysis on ACDC-snow.

