# OpenReview forum: "A Critical Look at Classic Test-Time Adaptation Methods in Semantic Segmentation"
_ICLR.cc/2024/Conference — ICLR 2024 Conference Withdrawn Submission_

### Official Review · Reviewer_U5Rv · 2023-10-30

**Soundness:** 2 fair
**Presentation:** 1 poor
**Contribution:** 1 poor
**Rating:** 3
**Confidence:** 3

**Summary:**

The paper anaylses common test-time adaptation (TTA) methods that adapt batch normalization (BN) statitstics of a model at test time;  for the task of semantic segmentation. The paper shows that TENT cannot really align BN statistics for the case of training on Cityscapes and testing on ACDC, a test set with distribution shifts. Then, it tries to rectify this behaviour by distributional estimation tricks like using a larger batchsize, or batch renormalization. Still, performance is lower than no TTA. Same when using a taecher-student model - for this specific trasnfer, no gains. Then the class imbalance problem is studied for the same usecase.

**Strengths:**

The paper deals with a learning setup, TTA, that has been shown to help performance for cases where test-time distribution shifts are observed.

**Weaknesses:**

W1: The paper presents a study on TTA for segmentation. Yet the analysis is on a single scenario (cityscapes -> ACDC/rain/fog) and the insights minimal. It is hard to understand if this behaviour also generalizes in more cases. I would expect at least one or two more source/test datasets and domains. In its current version, and without any technical contribution, as a study, this paper is weak.

W2: The experiments in Fig 1c and 1d are not adequately presented. Specifically, 1c the experiments with LN/GN are highly unclear, while the description of 1d is very hard to understand. The text would need to be rewritten, explained clearly and perhaps with way more details in an appendix.

W3: Section 4 is also written in a confusing way, without giving proper background for the experiment in Table 2. Also, from Figure 2 it is not easy to compare single vs TS performance  as one has to contrast nubmers from two tables. a plot would be far clearer

W5: the class imbalance analysis in sec5 is also not clear to me - it is a known fact that performance in the tail classes is lower and the results on how does adaptation affect that performance are unclear from fig 5.

**Questions:**

Q1: What are the lines for LN and GN in Fig1c? Is it the same network but with GN/LN instead of Batchnorm? is it trained in that regard? This experiment is unclear to me.

---

### Official Review · Reviewer_MvYz · 2023-11-02

**Soundness:** 2 fair
**Presentation:** 3 good
**Contribution:** 2 fair
**Rating:** 5
**Confidence:** 4

**Summary:**

This paper investigates the current test-time-adaptation (TTA) methods in semantic segmentation. The major finding is that current TTA methods for classification problem do not work well for semantic segmentation. Besides, the long-tailed imbalance issue for TTA segmentation is more severe than the issue for TTA classification.

**Strengths:**

- This paper thoughtfully investigates current TTA segmentation methods
- The paper is easy to follow
- The three findings of current TTA segmentation issues are somehow useful for the readers.

**Weaknesses:**

- Though the three findings are useful for potential readers, these findings are not surprising.
- The insightful analysis  related to the three issues about TTA segmentation is missing.
-Though the authors point out the three issues, no potential of investigation direction is given. Therefore, the novelty of this paper is somehow limited to me.

**Questions:**

Through this detailed investigation, what is the potential direction to explore for TTA segmentation?

---

### Official Review · Reviewer_GYgc · 2023-11-07

**Soundness:** 3 good
**Presentation:** 3 good
**Contribution:** 2 fair
**Rating:** 5
**Confidence:** 5

**Summary:**

Recently, test-time adaptation (TTA) has been widely studied in classification tasks. This paper systematically evaluated the applicability of traditional TTA methods on more challenging segmentation tasks and concluded some critical observations. For example, both batch norm updating strategy and teacher-student architecture cannot offer marked advantages for segmentation tasks. Additionally, severe long-tailed biases in segmentation also present a considerable obstacle. Extensive experimental and analytical results can provide some insights for the community.

**Strengths:**

### Originality:
This work stands out for its systematic exploration and comprehensive results, shedding light on the application of established TTA classification methods to segmentation tasks. The noteworthy insights regarding BN updating, the teacher-student EMA scheme, and long-tailed issues in segmentation have the potential to inspire the research community.

### Quality:
The authors present a rigorous and extensive experimental assessment of existing TTA classification techniques across three widely recognized semantic segmentation benchmarks. Each observation is accompanied by relevant techniques, detailed experimental outcomes, clear visualizations, and insightful explanations for the observed phenomena.

### Clarity:
This paper excels in its clarity and organization. It provides ample background information and effectively positions its contributions within the existing literature on test-time adaptation and semantic segmentation. Each section, spanning from Section 3 to Section 5, focuses on a key observation and thoroughly elaborates on the underlying reasons.

### Significance:
This work aims to inspire newcomers and engineers to focus on a more practical and challenging task of segmentation TTA, which has many applications in autonomous driving and robotics.

**Weaknesses:**

### Insufficient Evaluation:
The paper investigates the effectiveness of the BN updating strategy in TTA classification tasks, predominantly within CNN-based architectures. However, it deploys Segformer-B5, a Transformer-based architecture, for segmentation TTA tasks, which employs fewer BN layers. This choice raises the question of whether the use of Transformer-based architectures, rather than the segmentation task itself, may contribute to the potential ineffectiveness of the BN updating strategy. It would be beneficial for the authors to provide more comprehensive insights into this matter.


### Insufficient Observations:
The paper discusses patterns observed in classification tasks but does not adequately test these patterns against continuous (Wang et al., 2022) or persistent distribution shifts (Yuan et al., 2023). It is essential to expand the analysis beyond the context of fully test-time adaptation (Wang et al., 2021) to draw more meaningful conclusions regarding the broader applicability of the proposed strategies.


### Minor Issues:

- Some figures and the text within them, such as Figure 2 and Figure 3, appear too small, which can hinder the readability of the paper. I recommend that the authors include more detailed images or visualizations in the supplementary material to enhance the clarity of their work.

- It appears that there might be a typo error in the statement "while the count of class 14 is $10^5$." It should be corrected to "while the count of class 19 is $10^5$."

**Questions:**

### Fig. 1 (d) Performance Improvement and Ground-Truth Labels:
Fig. 1 (d) demonstrates a significant performance boost at a ground-truth label budget of 0.1%. This prompts two questions. Firstly, it is crucial to elaborate on the methodology and criteria for selecting and integrating these ground-truth labels into the segmentation tasks. Secondly, the paper should explore the potential performance trends if the budget were further reduced, as it appears to plateau.


### Batch Size between Segmentation and Classification:
Notably, the batch size used for segmentation tasks (ranging from 1 to 10) is still smaller than that employed for classification tasks (more than 64). To establish the efficacy of the BN updating strategy, additional evidence may be necessary to ascertain whether the current batch sizes are optimally utilized for segmentation tasks.


### Suggestions and Future Directions:
The paper would benefit from an expanded discussion of potential suggestions and future directions. This could encompass establishing a standardized baseline or benchmark for segmentation TTA research, fostering comparability and progress in this domain.

---

### Official Review · Reviewer_51SV · 2023-11-07

**Soundness:** 3 good
**Presentation:** 2 fair
**Contribution:** 1 poor
**Rating:** 3
**Confidence:** 4

**Summary:**

This paper focuses on the test-time adaptation for semantic segmentation. The authors conduct a systematic, empirical study about the challenges of TTA segmentation tasks and examine whether classic TTA strategies used in classification tasks are also applicable to segmentation tasks.

**Strengths:**

The analysis is meticulous, and the paper is well-written, making readers simple to grasp the fundamental concept.

**Weaknesses:**

1. The authors present a visual analysis of how mean and variance changes during online adaptation, but fail to explain why their results do not align with the true distribution of test data. Furthermore, the authors should also provide visual analysis for aligning batch norm statistics in classification tasks.
2. Similarly, the authors provide experiments that demonstrate the advantage of teacher-student models when pseudo-labels are inaccurate and find that they encounter bottlenecks when pseudo-labels quality are enhanced. However, they do not provide a more compelling, task-specific analysis for segmentation and do not provide a inspirational solution.
3. The details for data augmentation of ACDC-fog are missing.
4. As mentioned in the related work section, there has been some work focused on TTA segmentation such as [1].More discussions and analysis about these task-specific methods should be provided.


[1] Dynamically Instance-Guided Adaptation: A Backward-Free Approach for Test-Time Domain Adaptive Semantic Segmentation, CVPR2023

**Questions:**

Please see weaknesses section.

---

### Official Review · Reviewer_dCev · 2023-11-07

**Soundness:** 2 fair
**Presentation:** 2 fair
**Contribution:** 2 fair
**Rating:** 3
**Confidence:** 4

**Summary:**

The paper looks at test-time adaptation (TTA) where a model trained on a certain distribution is expected to adapt to a different distribution during test-time, without any supervision or labeled data. While a lot of work has been done for TTA on the image classification setting, this paper focuses on the image segmentation task, and analyzes popular TTA methods on image classification tasks like BN [6] and TENT [7] and shows that they do not have expected performance on segmentation task. The paper then goes on to identify some particular behavior of TTA methods on segmentation task, including how the estimation of batch norm statistics evolve during test-time, teacher student schemes and long-tailed problem in the segmentation task, and argues that further study into this direction would be important.

**Strengths:**

1. The paper is easy to read, and presents its ideas clearly. The flow of chapters is also natural.
2. I like the idea of the paper in that it tries to understand the shortcomings of image classification TTA methods for image segmentation, and has interesting study about multiple failure modes. I think this type of study can be helpful for the community to know what works/doesn’t work and why.
3. It is also important to shine more light on the segmentation TTA task, which I agree with the authors, is a bit under-studied compared to the classification case.

**Weaknesses:**

**(Improvement in writing)**

The writing of this paper could be improved. For example, the following segment is copied three times, with very little changes, in the abstract and the introduction, and two times within the introduction itself, providing very little extra information.

> Our comprehensive results have led to three key observations. First, the classic batch norm updating strategy, commonly used in classification TTA, only brings slight performance improvement, and in some cases it might even adversely affect the results. Even with the application of advanced distribution estimation techniques like batch renormalization, the problem remains unresolved. Second, the teacher-student scheme does enhance training stability for segmentation TTA in the presence of noisy pseudo-labels. However, it cannot directly result in performance improvement compared to the original model without TTA. Third, segmentation TTA suffers a severe long-tailed imbalance problem, which is substantially more complex than that in TTA for classification.

Also, adding background information about how the TTA methods work, in more detail, in the appendix would make the paper more self-contained. More information about the segmentation tasks/pretrained architecture in the appendix would also help.


**(Misses literature on segmentation TTA)**

The paper discusses test-time adaptation for semantic segmentation tasks, and shows that some popular TTA methods for image classification can be inadequate for the segmentation task. However, the paper fails to mention/analyze work done specifically for segmentation TTA. While it is possible that classification TTA methods are not sufficient for segmentation TTA, if there are specific methods for segmentation TTA, not discussing them makes the paper weaker.

I will mention a few examples here, but the list is not exhaustive.
1. SITA [1]. Looking at table 2 of this paper, it does seem TENT often does not perform well for segmentation. But SITA shows impressive numbers in this task. It also uses a batch size of 1 (single test instance).
2. [2] shows an interesting way of doing TTA for segmentation on night-time images using thermal information.
3. MM-TTA [3] shows interesting use of multi-modal information for 3D semantic segmentation.
4. Slot-TTA [4] uses a semi-supervised slot-centric scene decomposition model that at test time is adapted per scene through gradient descent on reconstruction or cross-view synthesis objectives.
5. Finally, [5] also works on how to adapt a segmentation model given a single unlabeled image without any other information.

**(Possible improvements in experimental results)**

For methods related to batch norm updating (Table 1), only TENT [7] and its variants are considered. There are other methods related to this strategy, for example BN [6], that have not been included in the table. **I understand that BN is discussed in a later section for a separate context, but seeing how they compare to source-only (SO) and TENT on segmentation tasks can be important.** Especially because comparison tables in SITA [1] show that BN performs better than TENT in segmentation tasks, so this feels like an incomplete/cherry-picked table.

**(Other segmentation models)**

The paper uses Segformer-B5 [9] as the pre-trained segmentation model. Experiments with more segmentation models will be important to establish the broad claims that this paper makes. We do not know if the results we are seeing are artifacts of this particular architecture/pre-training data, or are something more general. How about FAIR’s segment-anything [8] model/some other pretrained models?

**(Suggestion for overall improvement)**

While I appreciate the overall idea of this paper, i.e., analyzing failure of classification TTA methods on segmentation task, it feels incomplete without analyzing why certain segmentation TTA methods work well/what is missing from classification TTA methods. For example, if the paper shows the dynamics of SITA [1] or any of the other methods, show why it is performing well on segmentation tasks and what is different in them from image classification TTA methods, that would improve the paper vastly. SITA also seems to work well on classification tasks, so I feel it might not be the case that classification TTA methods just don’t work well on segmentation TTA tasks, rather something more interesting.

**Questions:**

**(Questions on figure 1)**

1. Are the training and test mean/variance in figure 1 over the entire train/test dataset? I imagine that is the case since they are horizontal lines.
2. For the test dataset statistics, do you train a model, from scratch, on the test dataset, and then use that model to calculate the test dataset statistics? Or do you use a model trained on the train dataset from scratch for this?
3. I am not certain about the difference between TENT and TENT (BN-initialized) in figure 1. For regular TENT, do you throw away the BN information from test data? Also not sure why TENT and TENT (BN-initialized) curves being close to each other is a surprising thing: if I understand correctly, the statistics are updated via exponentially moving averages, in which case the effect of initialization should go down as time progresses. One thought experiment: could you initialize the BN-statistics randomly, instead of using the statistics from the source model for initialization, and see if that changes anything? Similarly for TENT (regular), instead of calculating the statistics independently on each minibatch, could you use EMA to keep track of the statistics from prior mini-batches?
4. How susceptible are plots in figure 1 with the temporal order of images they see? I.e., if you randomize the order of images, would these plots change? See for example TTA against temporal correlation [10], which can be important when the TTA method is not looking at each example independently.
5. Why does batch estimation of variance differ so much from both train and test variance? It also starts below both train and test variance (even for TENT, BN-initialized) which does not make sense, and then estimating variance based on individual batches should not be so far away from the true variance, i.e., variance on the test data (red horizontal line). I am doubtful of these results.
6. Is it possible to generate plots similar to figure 1, but for a few image classification tasks? I could not find similar plots in the original TENT paper, but if any paper has that, citing that would also be fine. I am curious if this phenomenon is segmentation specific, or happens in other cases as well.

(**Question on figure 5**)
1. What is the definition of precision/recall for the segmentation task? Isn’t it possible to only be able to segment parts of an object, and not the full object? How is precision/recall measured in those cases?

> the reason might be that the long-tailed phenomenon may cause error accumulation at the pixel level and negatively affect the training process.

2. I feel more direct validation of the above reasoning/claim is necessary.


**(References)**

[1] SITA: Single Image Test-time Adaptation, https://arxiv.org/abs/2112.02355

[2] Test-Time Adaptation for Nighttime Color-Thermal Semantic Segmentation, https://arxiv.org/abs/2307.04470

[3] MM-TTA: Multi-Modal Test-Time Adaptation for 3D Semantic Segmentation, https://openaccess.thecvf.com/content/CVPR2022/papers/Shin_MM-TTA_Multi-Modal_Test-Time_Adaptation_for_3D_Semantic_Segmentation_CVPR_2022_paper.pdf

[4] Test-time Adaptation with Slot-Centric Models, https://arxiv.org/abs/2309.14052

[5] Single Image Test-Time Adaptation for Segmentation, https://arxiv.org/abs/2309.14052

[6] Improving robustness against common corruptions by covariate shift adaptation, https://arxiv.org/abs/2006.16971

[7] Tent: Fully Test-time Adaptation by Entropy Minimization, https://arxiv.org/abs/2006.10726

[8] Segment Anything, https://ai.meta.com/research/publications/segment-anything/

[9] Segformer: Simple and efficient design for semantic segmentation with transformers, https://arxiv.org/abs/2105.15203

[10] NOTE: Robust continual test-time adaptation against temporal correlation, https://arxiv.org/abs/2208.05117